# First Molecular Identification of Zoonotic *Babesia odocoilei* in Ticks from Romania

**DOI:** 10.3390/microorganisms13061182

**Published:** 2025-05-22

**Authors:** Ioan Cristian Dreghiciu, Diana Hoffman, Simona Dumitru, Ion Oprescu, Mirela Imre, Tiana Florea, Anamaria Plesko, Vlad Iorgoni, Sorin Morariu, Gheorghe Dărăbuș, Marius Stelian Ilie

**Affiliations:** 1Department of Parasitology and Parasitic Disease, Faculty of Veterinary Medicine, University of Life Sciences “King Mihai I” of Timisoara, 119, Calea Aradului, 300645 Timisoara, Romania; diana.hoffman@usvt.ro (D.H.); ionoprescu@usvt.ro (I.O.); mirela.imre@usvt.ro (M.I.); tijana.florea@usvt.ro (T.F.); plesko.anamaria.fa@usvt.ro (A.P.); sorinmorariu@usvt.ro (S.M.); gheorghedarabus@usvt.ro (G.D.); 2Veterinary and Food Safety Directorate 4, Surorile Martir Caceu, 300585 Timisoara, Romania; simonagiubega@gmail.com; 3Department of Infectious Diseases and Preventive Medicine, Faculty of Veterinary Medicine, University of Life Sciences “King Mihai I”, 300645 Timisoara, Romania; vlad.iorgoni@usvt.ro

**Keywords:** *Babesia odocoilei*, tick-borne diseases, zoonotic pathogens, *Dermacentor marginatus*, Romania, molecular identification, *Babesia* spp.

## Abstract

*Babesia odocoilei* is an emerging zoonotic protozoan parasite primarily associated with cervids, with growing recognition among non-cervid hosts and in terms of potential public health implications. While this species has been documented in North America and parts of Europe, data on its presence in Romania remain scarce. This study aimed to investigate the presence of *Babesia* spp. in ticks collected from Romania, providing new information on the existing species and their distribution, as well as their potential epidemiologic significance. A total of 41 *Ixodidae* ticks were collected from 184 wild boars across six counties from Western and Central Romania. Ticks were identified using morphological assessments, and DNA was extracted from the samples using a standardized protocol. The presence of *Babesia* spp. was assessed using real-time PCR with primers and a Taq Man probe targeting 116 bp fragments of 18S rRNA genes. Molecular analysis was used to detect *Babesia* spp. DNA from a single tick sample (1/41, 2.43%), identified as *Dermacentor marginatus*, from Timiș County. The resulting amplicons were sequenced and compared with reference sequences in GenBank for species confirmation. This finding represents the first molecular identification of *B. odocoilei* in questing ticks from Romania. The expanding host range and geographic distribution of *B. odocoilei* raise concerns regarding its zoonotic potential. The presence of this pathogen in *Dermacentor marginatus* ticks suggests a broader vector competence than previously recognized, and future research should focus on host reservoirs, vector competence, and potential zoonotic transmission, with an emphasis on public health implications, including potential implications for veterinary diagnostics, vector control policies, and public health awareness regarding emerging tick-borne pathogens.

## 1. Introduction

Tick-borne zoonotic pathogens represent a significant and emerging public health threat across Europe. The increasing prevalence of tick-borne diseases (TBDs) has been attributed to the complex interplay of environmental, ecological, and socio-political factors, including climate change, habitat modifications, wildlife population dynamics, and human activities [1,2]. These factors collectively influence the geographic distribution, density, and activity of tick populations, thereby facilitating the transmission of tick-borne pathogens to humans and animals [1].

Among the various TBDs, babesiosis is an important yet often underdiagnosed zoonotic disease caused by intraerythrocytic protozoan parasites of the genus *Babesia*. These parasites are transmitted primarily by ixodid ticks and infect a wide range of vertebrate hosts. In humans, *Babesia* infections can lead to a spectrum of clinical manifestations, ranging from asymptomatic cases to severe, life-threatening hemolytic anemia, particularly in immunocompromised individuals [3,4].

The transmission cycle of *Babesia* spp. involves a complex network of vertebrate hosts and tick vectors. The maintenance of these parasites in nature is largely dependent on their interactions with specific reservoir hosts and vector species. Various *Babesia* species have evolved distinct host preferences and transmission strategies, with some primarily maintained within wildlife reservoirs. [5]. The ongoing changes in wildlife migration patterns, largely driven by environmental and climatic shifts, have further contributed to the expansion of infectious disease ranges, altering the epidemiology of babesiosis in Europe and beyond [6,7,8].

The first documented observation of *Babesia* spp. dates to 1888 when Victor Babeș identified intracellular, round, and frequently paired microorganisms within the erythrocytes of febrile cattle suffering from hemoglobinuria [9]. Since this discovery, over 100 species of *Babesia* have been described, infecting a diverse range of wild and domestic animals [10,11]. While most *Babesia* spp. are primarily of veterinary importance, a limited number have been identified as zoonotic agents capable of infecting humans [11].

Based on classical taxonomy, piroplasms are broadly classified into three distinct groups: the *Theileria* genus, which includes *Theileria capreoli* (Clade V [4]), *Babesia* (sensu stricto) group, comprising species such as *Babesia canis*, *Babesia venatorum*, *Babesia odocoilei*, *Babesia divergens*, and *Babesia capreoli* (Clade VI, [4]), and the *Babesia* (sensu lato) group, which includes *Babesia microti* (Clade I, [4]) [12].

This classification system has been instrumental in understanding the evolutionary relationships and host–parasite interactions within the *Babesia* genus. However, advancements in molecular phylogenetics continue to refine our understanding of *Babesia* spp., their genetic diversity, and their pathogenic potential [11,13]. Notably, ixodid ticks serve as the primary vectors for the majority of *Babesia* spp., facilitating their transmission through blood-feeding on vertebrate hosts. Although *Babesia meri* is an exception, being transmitted by the argasid tick *Ornithodoros erraticus* [14], all other known *Babesia* species are vectored exclusively by ixodid ticks [15]. The presence of multiple pathogen species within tick populations has raised concerns regarding co-infections, where *Babesia* spp. may be transmitted alongside other tick-borne pathogens, such as *Borrelia burgdorferi* (the causative agent of Lyme disease), *Anaplasma* spp., and *Ehrlichia* spp. [16,17]. These co-infections can result in more severe clinical outcomes in both human and animal hosts, necessitating a comprehensive approach to the diagnosis, prevention, and management of TBDs [18].

In Hungary, *Babesia canis* DNA was detected in four *Dermacentor reticulatus* ticks (2.4%; 4/165). One *Haemaphysalis concinna* nymph (3.4%; 1/29) was found to carry *Babesia* cf. *crassa* DNA, showing 100% sequence identity with a zoonotic strain previously reported in Murska Sobota, Slovenia—approximately 80 km from the sampling site elected for the present study. A phylogenetic analysis placed this isolate within the Far-Eastern *Babesia* genotype group, commonly associated with *H. concinna*. No piroplasm DNA was detected in *Ixodes ricinus*, *D. marginatus*, or *H. inermis*, suggesting species-specific vector associations for these pathogens [19].

Certain biological families, including cervids and bovids, play a crucial role in the epidemiology of babesiosis by acting as reservoir hosts for *Babesia* spp. Among these, *Babesia odocoilei* has been identified in both feral and captive populations of white-tailed deer (*Odocoileus virginianus*), highlighting the importance of wildlife in the persistence and transmission of the parasite [20,21]. The increased interface between wildlife and human populations, coupled with anthropogenic environmental changes, has contributed to the emergence and spread of babesiosis in previously unaffected regions [22].

The first confirmed case of human babesiosis in Europe was reported in Croatia in 1957 [23]. Since then, the incidence of human babesiosis has shown a rising trend, with increasing case numbers reported across multiple European countries [17,24]. In Slovakia, three human cases have been recorded since 1991, indicating the ongoing expansion of babesiosis within the region [25].

The elevated prevalence of *Babesia* infections among individuals with a history of tick bites and tick-borne diseases suggests the potential co-transmission of multiple pathogens mediated by infected ticks. Occupational exposure among foresters, farmers, veterinarians, and herders likely contributes to the heightened risk in these populations. Notably, the detection of zoonotic *Babesia* species in blood donors raises concerns about transfusion-associated babesiosis—a growing issue in endemic regions, with documented cases in the United States and elsewhere [26,27]. Additionally, asymptomatic carriers, particularly travelers moving from endemic to non-endemic areas, may facilitate inadvertent transmission and misdiagnosis, while serving as reservoirs for local tick populations [28]. Although often asymptomatic in immunocompetent individuals, babesiosis poses significant risks for severe or fatal outcomes in those with underlying conditions, such as splenectomy, HIV infections, malignancies, or old age [29,30,31]. These findings underscore the need for enhanced surveillance, diagnostic awareness, and preventive strategies, particularly in blood transfusion services and among high-risk groups.

Over the last three decades, the number of reported human babesiosis cases has steadily increased. This rising trend is attributed to multiple factors, including an actual increase in incidence, enhanced diagnostic capabilities, and heightened public health awareness. Despite being historically considered a rare disease, babesiosis is now recognized as an emerging zoonotic threat, necessitating improved surveillance and prevention strategies [17,24].

This study aimed to investigate the presence of *Babesia* spp. in ticks collected from Romania, providing new information on the existing species and their distribution, as well as their potential epidemiologic significance. By characterizing the prevalence and distribution of these pathogens, the study seeks to contribute to the growing body of knowledge on TBDs and their impact on both animal and human health.

## 2. Materials and Methods

### 2.1. Sample Collection

Hard ticks were collected from wild boar tails during the period October 2021–February 2024 from hunting grounds located in six counties from Western and Central Romania, namely Timiș, Caraș-Severin, Hunedoara, Alba, Sibiu, and Mureș.

The sampling counties are displayed in Figure 1. Tick specimens were morphologically identified following established criteria [32] and subsequently preserved in 70% ethanol.

### 2.2. DNA Extraction

The ticks were homogenized in the tissue homogenizer at 4000 m/s for 10 min in 1.4 mm diameter ceramic bead tubes together with 500 µL of PBS (phosphate-buffered saline) (Figure 2). Prior to the extraction process, 45 ticks were washed several times with ultrapure water and dried on filter paper. However, during the dissection process, four specimens were excluded from further analysis due to excessive destruction, which compromised tissue viability, or due to a poor morphological condition suggestive of degradation or contamination. These factors rendered them unsuitable for molecular processing. Consequently, 41 ticks were deemed viable and included in the study for the downstream examination and analysis (Figure 3). For better homogenization and release of the contents, the ticks were sectioned into two or four parts, depending on their size, in a Petri dish with a sterile scalpel blade for each sample. A mechanical lysis was performed in order to break the chitinous envelope and release the contents.

After homogenization, they were centrifuged at 13,000 rpm for 3 min, and the amount of supernatant taken into work was 200 µL.

The MagMax^TM^ Core Nucleic Acid Purification Kit (Applied Biosystems^TM^, Waltham, MA, USA) and the KingFisher^TM^ Flex Purification System (Thermo Scientific^TM^, Waltham, MA, USA) were used for DNA/RNA extraction, according to the manufacturer’s instructions.

### 2.3. Molecular Analysis of Ticks for Babesia spp.

The samples were subjected to the protocol described by Stańczak, 2015 [34] with a TaqMan^®^ MGB^TM^ (ThermoFisher Scientific, Pleasanton, CA, USA) (minor groove binder) probe, Bab18S-p FAM-AAGTCATCAGCTTGTGCAGATTAC GTCCCT-BHQ1 (3 μM), and a pair of primers (Bab18S-f CATGAACGAGGAATGCCTAGT ATG and Bab18S-r CCGAATAAT TCA CCG GAT CAC TC) (10 μM). The probe and primers were designed to detect a 116 bp fragment in the 18S rRNA gene.

The amplification was carried out using a QuantStudio^TM^ 7 Flex Real-Time PCR System (Applied Biosystems^TM^, Pleasanton, CA, USA) and after each sample was tested, the Ct value was determined based on the log-linear phase of each reaction. The results were interpreted on the basis of a standard Ct relative to the number of amplification cycles, and the threshold was automatically selected by the PCR platform to not influence the result.

Prior to the amplification, the primers, probe, and master mix were tested with a positive control—a positive sample of *Babesia caballi*, amplified and sequenced in a previous study (PQ317203.1) [35].

The following additional controls were added to each reading cycle for plate validation: ME (negative extraction control—water or a known safe negative sample) and NTC (no template control—amplification mix + ultrapure water).

### 2.4. Conventional PCR

For sequencing, the positive sample was tested via conventional PCR. The protocol used 5 μL of DNA, 12.5 μL of MyTaqTM Red Mix (BIOLINE^®^ Ltd., London, UK), 1 μL of the primer F, 1 μL of the primer R, and 5.5 μL of ultrapure water. Amplification was carried out with a My Cycler thermocycler (Bio-Rad^®^, Berkeley, CA, USA) via a 32-cycle amplification program, i.e., initial DNA denaturation at 95 °C for 1 min, followed by denaturation at 95 °C for 30 s, alignment at 58 °C for 30 s, and extension at 72 °C for 30 s, then followed by incubation at 4 °C. Amplicon analysis was performed via horizontal electrophoresis in a 1.5% agarose gel submersion system with the addition of the fluorescent dye RedSafe^TM^ (iNtRON Biotechnology, Inc., Seongnam-si, Gyeonggi-do, Republic of Korea), at a voltage of 120 V and 90 mA, for 60 min. The 100 bp DNA ladder marker (BIOLINE^®^ UK Ltd., London, UK) was used in the first well of the gel. After migrating the samples in the agarose gel, the image of the gel with the migrated DNA fragments was captured using a UV photo documentation system (UVP^®^, Upland, CA, USA).

In order to identify the species of *Babesia* spp., the amplicons were purified using the α + Solution^TM^ GEL/PCR Purification Kit (Alphagen, Changzhi, Taiwan), according to the manufacturer’s recommendations, and sent for sequencing (Sanger sequencing) in the forward and reverse direction by the company Macrogen^®^ Europe B.V., Amsterdam, The Netherlands. The obtained sequences were edited, and a homology search was performed using the online version of the Basic Local Alignment Search Tool (BLAST) and compared with those available in the Gen-Bank database (https://blast.ncbi.nlm.nih.gov, accessed on 14 March 2025) [36].

## 3. Results

### 3.1. Tick Infestation of Wild Boars

A total of 41 *Ixodidae* ticks were collected from 184 examined wild boars, with some animals hosting multiple ticks, from different hunting grounds located in six counties from Romania. Five species of ticks belonging to three genera were found, namely *Ixodes ricinus*, *Dermacentor marginatus*, *Dermacentor reticulatus*, *Haemaphysalis concinna*, and *Haemaphysalis erinacei* (Table 1).

### 3.2. Molecular Detection of Babesia spp. in Ticks

Out of 41 analyzed tick DNA samples, *Babesia odocoilei* was detected in a single sample, corresponding to a prevalence of 2.43%. The cycle threshold (Ct) value for this positive sample, as presented in Figure 4, was 23.067. This sample (Sample 41) was obtained from a tick identified as *D. marginatus* from Timiș county, the largest county in Romania from a territorial point of view, with 8,696.7 km^2^ (3.65% of the country’s surface area), encompassing all landforms, with altitudes ranging from 75 to 1384 m. The climate is temperate continental with Mediterranean and oceanic influences [37]. Conventional PCR yielded positivity for the 116 bp fragment in the 18S rRNA gene.

### 3.3. Sequencing Results

The results of the sequencing revealed a *Babesia odocoilei* species with high similarity to several GenBank isolates, such as MH899097.1 (Canada), AY661510.1 (Texas), and KC460321.1 (Canada).

The sequence was deposited into GenBank with the accession number PV156327.1.

## 4. Discussion

*Babesia odocoilei*, a protozoan parasite primarily associated with American white-tailed deer (*Odocoileus virginianus*), has been documented as a causative agent of babesiosis in cervids and, in some instances, in bovid species in Europe [38,39]. In the present study, we identified *B. odocoilei* in a questing *I. ricinus* nymph, marking the notable detection of this parasite in an active host-seeking tick. This finding is particularly significant given the expanding recognition of *B. odocoilei* beyond its previously well-established host range [40].

Previous studies have reported the presence of related genotypes, such as *Babesia* cf. *odocoilei*, in *I. ricinus* ticks across Europe [14,41]. However, despite their genetic similarity to *B. odocoilei*, the zoonotic potential of these related *Babesia* lineages remains poorly understood. While *B. odocoilei* has been primarily associated with cervid hosts, the detection of these related genotypes in *I. ricinus*, a tick species with a broad host range that includes humans, raises important questions regarding their potential for human infection and their role in the epidemiology of tick-borne babesiosis [40].

Gandy et al., 2024, in a study conducted in England and Wales found a significant positivity rate for *Babesia* spp. and the first detection of *B. odocoilei*-like species in questing ticks from the United Kingdom, specifically in Exmoor (1/294) [42].

The presence of *B. odocoilei*-like species in moose (*Alces alces*) was first documented in Norway by Puraite et al., 2016, marking significant expansion of the known host range of this parasite. This detection was particularly noteworthy as it represented the first identification of *B. odocoilei*-like species in moose, a species not previously recognized as a reservoir for this pathogen [43].

The discovery of *B. odocoilei*-like species in moose suggests that this parasite may have a broader host range than previously assumed, with potential implications for wildlife ecology and disease-transmission dynamics. This finding also raises questions regarding the vector competence of tick species in Norway, as the transmission cycles of *Babesia* spp. are highly dependent on the presence of competent tick vectors. In addition to its widespread occurrence in free-ranging wildlife, epidemiological studies have reported instances of *B. odocoilei* causing fatal infections in bovid species under captive conditions [44,45].

The increasing recognition of *B. odocoilei* as a zoonotic pathogen of concern necessitates further research to elucidate its vector competence, host range, and potential risks to public and veterinary health. The widespread distribution of *Babesia odocoilei* across North America has been well-documented over the past half-century, with numerous studies confirming its presence in both tick vectors and vertebrate hosts. In the United States, the pathogen has been identified across a broad geographical range, extending from the northeastern state of Maine [46,47,48] to the southern regions of Texas [49], and as far west as California [44].

The detection of *B. odocoilei* in California is particularly noteworthy, as it has been found in desert bighorn sheep, a non-cervid vertebrate host. This observation is significant because it suggests that *B. odocoilei* can infect a broader range of mammalian hosts beyond its traditionally recognized cervid reservoirs. Furthermore, the pathogen’s presence in this region occurs outside the known distribution of *Ixodes scapularis*, the primary tick vector associated with its transmission [44].

In Romania, a total of 852 ticks belonging to ten different species were screened for apicomplexan parasites using molecular methods. *Babesia canis* was identified in seven samples, specifically in *Dermacentor reticulatus* (five samples) and *Ixodes ricinus* (two samples). A single *Babesia ovis* infection was detected in a *Rhipicephalus bursa* specimen collected from a sheep in Ciochiuta, while *Babesia microti* was found in two *I. ricinus* ticks—one from a fox in Corbeanca and another from a dog in Timișoara [50]. To date, only two other studies have reported the molecular detection of apicomplexan parasites in ticks in Romania. Paduraru et al. (2012) identified a single *Babesia venatorum* (formerly *Babesia* sp. EU1) sequence from 146 *I. ricinus* ticks collected from roe deer and goats [51]. Similarly, Ionita et al. (2013) screened 382 ticks from five species, primarily collected from cattle, and reported the presence of *B. microti* and *B. occultans* [52]. These findings highlight the diversity of *Babesia* species circulating among different tick vectors and host species in Romania and underscore the importance of continued molecular surveillance for tick-borne protozoan pathogens.

Given the complexity of *Babesia* spp. transmission and their potential to cause severe disease in humans and animals, further research is essential to elucidate their epidemiology, pathogenicity, and ecological drivers. The continued surveillance of *Babesia* spp. in tick populations and their reservoir hosts will be critical in assessing the risk of human infection and formulating effective mitigation measures [53].

Further investigations are required to determine whether *Babesia* cf. *odocoilei* and other related genotypes can infect humans or domestic animals. Molecular and epidemiological studies should aim to clarify the transmission dynamics, host specificity, and potential pathogenicity of these parasites. Given the increasing overlap among wildlife, domesticated animals, and human populations, understanding the zoonotic risk associated with *B. odocoilei* and its related species is crucial for public health and veterinary disease management.

Although this work provides the first DNA identification of *B. odocoilei* in Romanian ticks, it should be noted that it has a few limitations. First, it is challenging to draw firm conclusions regarding prevalence because there were only 41 ticks examined and only one sample was confirmed to be positive. Second, the existence of *B. odocoilei* in other regions cannot be ruled out because tick collection was restricted to six counties in Western and Central Romania. It is also yet unclear how wild boars could be involved as reservoir hosts. Finally, the public health consequences of this result are still uncertain because no clinical data from people or domestic animals were included.

## 5. Conclusions

These findings underscore the potential clinical significance of *Babesia odocoilei*, particularly in susceptible hosts, such as immunocompromised individuals or animals, and in ecosystems where tick–host interactions may be shifting due to environmental or anthropogenic changes. Notably, in contrast to data reported from Romania’s neighboring countries, our comprehensive review revealed a lack of prior studies investigating this specific pathogen within Romanian tick populations.

The primary novel contribution of this study is the molecular detection of *Babesia odocoilei* in ticks collected in Romania—an observation that, to the best of our knowledge, represents the first documentation of this piroplasm in local tick species. This finding expands the known geographic range of *B. odocoilei* and raises important questions about its epidemiological role, the potential risk for transmission to wildlife or domestic animals, and its broader implications for public and veterinary health.

Continued surveillance and targeted research are warranted to better understand the ecology, host range, and zoonotic potential of this emerging pathogen.

## Figures and Tables

**Figure 1 microorganisms-13-01182-f001:**
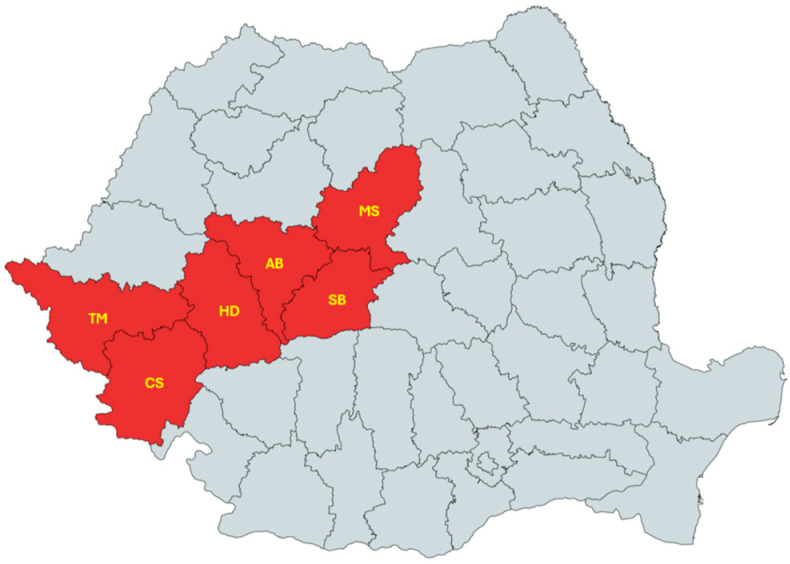
Map of Romania showing counties where ticks were collected from wild boars [33].

**Figure 2 microorganisms-13-01182-f002:**
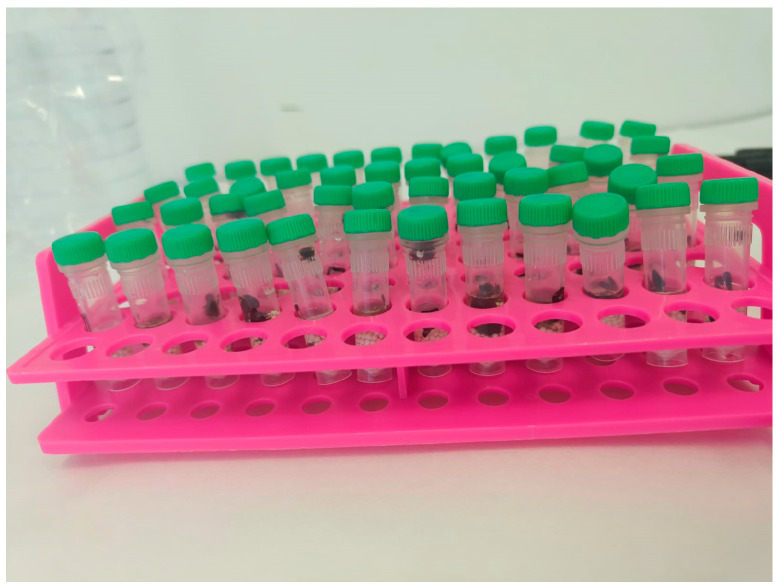
Ticks homogenized in tubes with ceramic beads.

**Figure 3 microorganisms-13-01182-f003:**
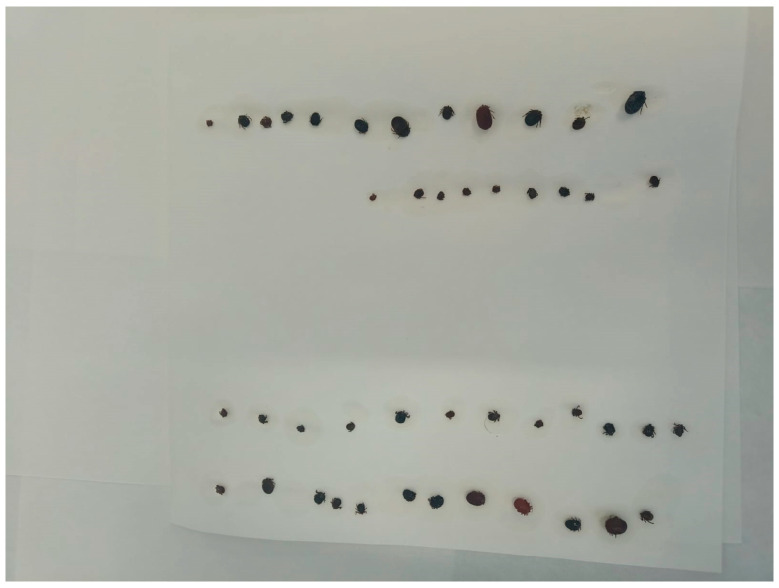
Dried ticks after they were washed in ultrapure water.

**Figure 4 microorganisms-13-01182-f004:**
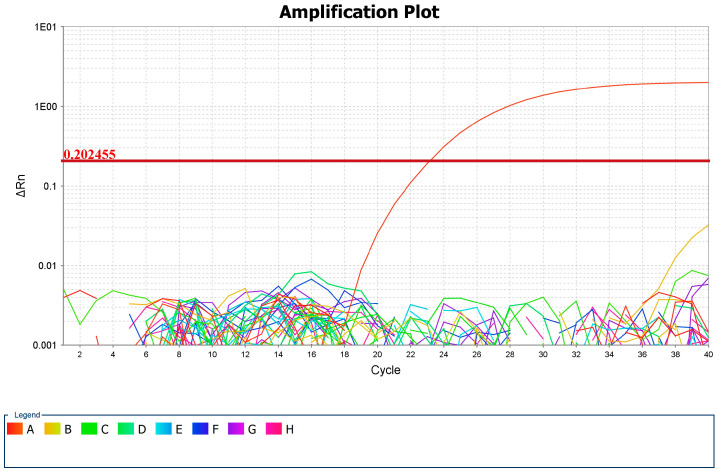
Positive sample log curve in the amplification plot and CT value (A—*B. odocoilei*, B, C, D, E, F, G, H—other negative samples).

**Table 1 microorganisms-13-01182-t001:** Epidemiological factors of identified ticks.

Epidemiological Factors	No. of Ticks
County	
Timiș	10
Caraș Severin	5
Hunedoara	7
Alba	5
Sibiu	6
Mureș	8
Gender	
Male	9
Female	32
Stage of development	
Adult	41/41
Larvae	0/41
Nymph	0/41
Species	
*I. ricinus*	8
*D. reticulatus*	13
*D. marginatus*	16
*H. concinna*	1
*H. erinacei*	2

## Data Availability

Data are contained within the article.

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
