# Peer review of "First Molecular Identification of Zoonotic Babesia odocoilei in Ticks from Romania"

_microorganisms, 2025, doi:10.3390/microorganisms13061182_

Round 1
Reviewer 1 Report
Comments and Suggestions for Authors
The authors describe methods used and report findings relative to the first molecular (RT-PCR) detection of Babesia odocoilei in hard ticks collected in Romania. Overall, the manuscript is well supported by appropriate referenced information and clearly details the detection and the potential human and veterinary health implications of their findings. Additional details related to the authors’ findings may enhance the impact of these findings and are listed below.
Introduction
Lines 89-92 – This paragraph seems out of place regarding the flow of the introduction section. The authors discuss taxonomic classification of piroplasms (lines 62-65), moves to discussion of vectors (lines 67-76) and hosts (animal and human) (lines 77-87), then begins discussing taxonomic classification again before discussing human cases again. Suggest moving paragraph (lines 89-92) to follow line 66.
General question – have there been other studies that surveyed for Babesia spp. in wildlife or ticks in Romania or neighboring countries? Such information would be useful to include in the introduction.
Materials and Methods
Line 106 – Are the months October – February the general tick season in Romania?
Lines 116 and 121 – Figures are not in correct order; Figure 3 was mentioned before figure 2.
Line 117 – It is mentioned the ticks were sectioned depending on their size. Was there any standardization on which ticks were cut into two and which were cut into four?
Line 128 – Figures are good but do not significantly add to/enhance the manuscript. Showing the beads or the homogenized ticks would be more useful in Figure 2. Figure 3 does not seem to be focused adequately to convey exactly what is being represented.
Lines 130-134 – Suggest authors include more information regarding the PCR protocol, such as specific information on the enzyme used, PCR conditions, etc.?
Line 134 – Suggest authors include exactly what the primer/probe are amplifying. Is it for general piroplasms, both Babesias, Babesia sensu stricto, Babesia sensu lato, etc.?
Line 140 – Was there a standardized Ct value cutoff for determination of positive samples?
Results
Section 3.1 – Suggest adding a table listing the exact number of ticks per species, life stages, and collection location.
Line 181 – One late sample (“B”) seems to be amplifying. Refer to the last comment in materials and methods. The amplification could be background noise, but that is difficult to determine if the reader has no Ct cutoff to reference.
Line 179 – Suggest authors briefly describe the ecology of Timis County, or have a figure to highlight its location.
Discussion
Line 231 – Any implications or previous findings of Babesia odocoilei in Dermacentor marginatus?
Good last paragraph as it provides a clear summary of the need for surveillance focused studies.
Conclusion
Lines 240-241 – Although the sentence is well formatted, the findings do not necessarily support what it is attempting to convey. There is no mention of examining or analyzing clinical severity, or if the host species had pathology associated with Babesia odocoilei infection.
Line 249 – Is there a more efficient way state the pathogen has not been found in neighboring countries.
Lines 249 – 252 – These can be combined to make one solid and well-rounded paragraph.
Reviewer 2 Report
Comments and Suggestions for Authors
Thank you for the review opportunity!
This paper focused on finding Babesia DNA within ticks in Western Romania. While the article is interesting i do have some questions and suggestions if the authors choose to adhere.
Abstract: please mention what's the abbreviation "D. marginatus" for.
Why only 41 ticks were included in the study ? The sample size seems rather small in my point of view.
The final part of the abstract should be focused on the practical implications of your findings in both animal health and human health, not general information applicable for any finding.
Introduction: line 41: please include a citation.
Line 61: please include a citation
I would add a paragragh on the implication of Babesia spp. in human health as well in the introduction section.
Figure 2 and 3 are unclear, they should be resized and enhanced.
Figure 3. I count 45 ticks, why were only 41 included in the study ?
Materials and methods are well written, with nothing much to add.
Results:
"A total of 41 Ixodidae ticks were taken from 184 wild boar" how is this even possible ? if it's 1 tick per boar the maximum is 41 ... Please explain and consider rephrasing.
Line 172 - 174: should be offered a citation and moved into the discussion section.
Discussion:
The discussion are well written but lack a limitation section and a paragraph on previous data reported in Romania. Please consider adding.
Conclusions:
These section should be expanded focusing more on your findings and on further implications it may have for both animals and humans.
I'm also not a big fan of including further research directions in the conclusion section. Consider moving into the discussion section.
Overall comments:
I consider the data offered to not be sufficient for a full-article; maybe consider to transform this manuscript into a "short communication". Otherwise, please explain the reasoning to present a whole article on only 41 ticks since they were just screened for Babesia spp. If more pathogens were included i could see why it should be an article.
The English is rusty at best and should be enhanced, please consider finding a native English speak to aid or take advantage of the English Editing Services offered by MDPI.
Also the manuscript seems to not be in its final form since it was sent with "Track changes".
Comments on the Quality of English LanguageShould be improved.
Round 2
Reviewer 2 Report
Comments and Suggestions for Authors
"A total of 45 ticks were initially collected and thoroughly washed with ultra-pure water, as illustrated in the image. However, during the dissection process, four specimens were excluded from further analysis due to excessive destruction, which compromised tissue viability, or due to poor morphological condition suggestive of degradation or contamination. These factors rendered them unsuitable for molecular processing. Consequently, 41 ticks were deemed viable and included in the study for downstream examination and analysis."
Please note this information should be added in the main manuscript as well.
"Thank you for your thoughtful evaluation. We understand your concern regarding the sample size and scope. While the number of ticks analyzed may appear limited, we believe the manuscript provides meaningful scientific value for several reasons. First, this is the first report of Babesia odocoilei in tick species in Romania, which represents a novel and significant finding with potential implications for both veterinary and public health. Second, the ticks were collected from wild boars—a host species increasingly recognized as a reservoir for tick-borne pathogens—highlighting an underexplored ecological interface. While the study focuses on Babesia spp., the detection of a zoonotic piroplasm in this context justifies broader attention."
Please note that in my initial comment i did not contest the importance of your findings and i completely agree with the authors on value of their work and the need to be published, even for international readers. My questions, which i feel they did not answer, is: why do they consider the data to be sufficient for a FULL ARTICLE and a BRIEF COMMUNICATION is not enough to report the findings considering only 41 ticks were analyzed and only one pathogen was determined ?
Limitations were not added, please consider.
I recommend acceptance of the article with minor revisions, after the authors address my last, few, concerns.
